

# Development and application of molecular biomarkers for characterizing Caribbean Yellow Band Disease in *Orbicella faveolata*

Michael Morgan[1], Kylia Goodner[2], James Ross[1], Angela Z. Poole[3], Elizabeth Stepp[4], Christopher H. Stuart[5], Cydney Wilbanks[1] and Ernesto Weil[6]

[1] Department of Biology, Berry College, Mount Berry, GA, United States
[2] Department of Genetics, Yale University, New Haven, CT, United States
[3] Department of Biology, Western Oregon University, Monmouth, OR, United States
[4] The Medical College of Georgia, Georgia Regents University, Augusta, GA, United States
[5] Department of Molecular Medicine, Wake Forest School of Medicine, Winston-Salem, NC, United States
[6] Department of Marine Sciences, University of Puerto Rico, Lajas, Puerto Rico, United States

Corresponding author
Michael Morgan,
mbmorgan@berry.edu

## ABSTRACT

Molecular stress responses associated with coral diseases represent an under-studied area of cnidarian transcriptome investigations. Caribbean Yellow Band Disease (CYBD) is considered a disease of *Symbiodinium* within the tissues of the coral host *Orbicella faveolata*. There is a paucity of diagnostic tools to assist in the early detection and characterization of coral diseases. The validity of a diagnostic test is determined by its ability to distinguish host organisms that have the disease from those that do not. The ability to detect and identify disease-affected tissue before visible signs of the disease are evident would then be a useful diagnostic tool for monitoring and managing disease outbreaks. Representational Difference Analysis (RDA) was utilized to isolate differentially expressed genes in *O. faveolata* exhibiting CYBD. Preliminary screening of RDA products identified a small number of genes of interest (GOI) which included an early growth response factor and ubiquitin ligase from the coral host as well as cytochrome oxidase from the algal symbiont. To further characterize the specificity of response, quantitative real-time PCR (qPCR) was utilized to compare the expression profiles of these GOIs within diseased tissues (visible lesions), tissues that precede visible lesions by 2–4 cm (transition area), and tissues from healthy-looking colonies with no signs of disease. Results show there are distinctive differences in the expression profiles of these three GOIs within each tissue examined. Collectively, this small suite of GOIs can provide a molecular "finger print" which is capable of differentiating between infected and uninfected colonies on reefs where CYBD is known to occur.

## INTRODUCTION

Worldwide, marine diseases are quickly spreading and creating a large ecological and economic problem for global marine ecosystems (*Harvell et al., 1999*; *Harvell et al., 2002*; *Burge et al., 2014*). This problem is particularly important in coral reef ecosystems which have experienced significant declines and phase shifts to algal dominated communities over the past 30 years. Correlations are known to exist between increasing sea water temperatures, extensive and intensive bleaching events, and the prevalence of disease outbreaks (*Bruno et al., 2007*; *Hoegh-Guldberg et al., 2007*; *Eakin et al., 2010*; *Ruiz-Moreno et al., 2012*). Yellow Band Disease is a disease affecting reef-building corals in the Caribbean and Indo-Pacific regions characterized by a distinct yellow-to-white wide (1–10 cm) band or halo pattern separating healthy-looking tissue and the algae-colonized, tissue-denuded skeleton (Fig. 1) of infected colonies (*Gil-Agudelo et al., 2004*; *Weil, 2004*) and apparently by similar pathogens (*Cervino et al., 2004a*; *Cervino et al., 2008*). The Caribbean is a disease hot spot (*Weil, 2004*) and Caribbean Yellow Band Disease (CYBD) has caused significant mortalities in some of the major reef-building genera (*Orbicella*, *Montastraea*) in the region since 1997 (*Gil-Agudelo et al., 2004*; *Bruckner & Bruckner, 2006*; *Bruckner & Hill, 2009*; *Weil, Cróquer & Urreiztieta, 2009a*; *Weil & Rogers, 2011*). Although no mode of transmission has been identified (*Weil et al., 2008*), it has been reported that CYBD is caused by infection of the zooxanthellae with at least three *Vibrio* spp. causing degradation to the symbiotic dinoflagellates of the genus *Symbiodinium* that reside within coral gastrodermal cells (*Cervino et al., 2004a*; *Cervino et al., 2004b*; *Cervino et al., 2008*; *Cunning et al., 2008*). This relationship forms the basis for the productivity and diversity of reef ecosystems and therefore understanding how this disease influences the holobiont is extremely important in mitigating the spread of this disease.

Unlike other coral diseases, during infection with CYBD, *Symbiodinium* exhibit compromised cellular integrity, loss of pigmentation and mortality. Algal cells remain inside the coral endoderm, but as coral tissue loses pigmentation and transitions from yellow to pale yellow to white, and is similar in appearance to bleached coral, most algal cells are dead (*Cervino et al., 2004a*; *Cervino et al., 2008*). The signs on the coral colonies are bands or halos of yellow-pale or bleached tissues bordering the dead tissue areas on one side and fringing healthy-looking tissue on the other (see Fig. 1) (*Cervino et al., 2004a*; *Gil-Agudelo et al., 2004*). Compounding the effects of the disease, rising global water temperatures allows *Vibrio* to thrive (*Harvell et al., 1999*; *Harvell et al., 2009*). Comparisons between healthy and diseased corals at slightly elevated water temperature found that while healthy corals survive, diseased corals had a 60–80% mortality rate within a 96-hour period (*Cervino et al., 2004a*; *Cervino et al., 2004b*). The disease has a systemic effect significantly reducing fecundity of infected colonies and therefore, fitness of populations and species reducing the potential for natural recovery (*Weil, Cróquer & Urreiztieta, 2009a*). As worldwide water temperatures continue to rise, conditions favor new infections and higher virulence of the *Vibrio* species that cause CYBD (*Weil, Cróquer & Urreiztieta, 2009a*). Therefore it is important to further clarify the transmission and progression mechanisms in order to manage the disease and protect Caribbean coral ecosystems and

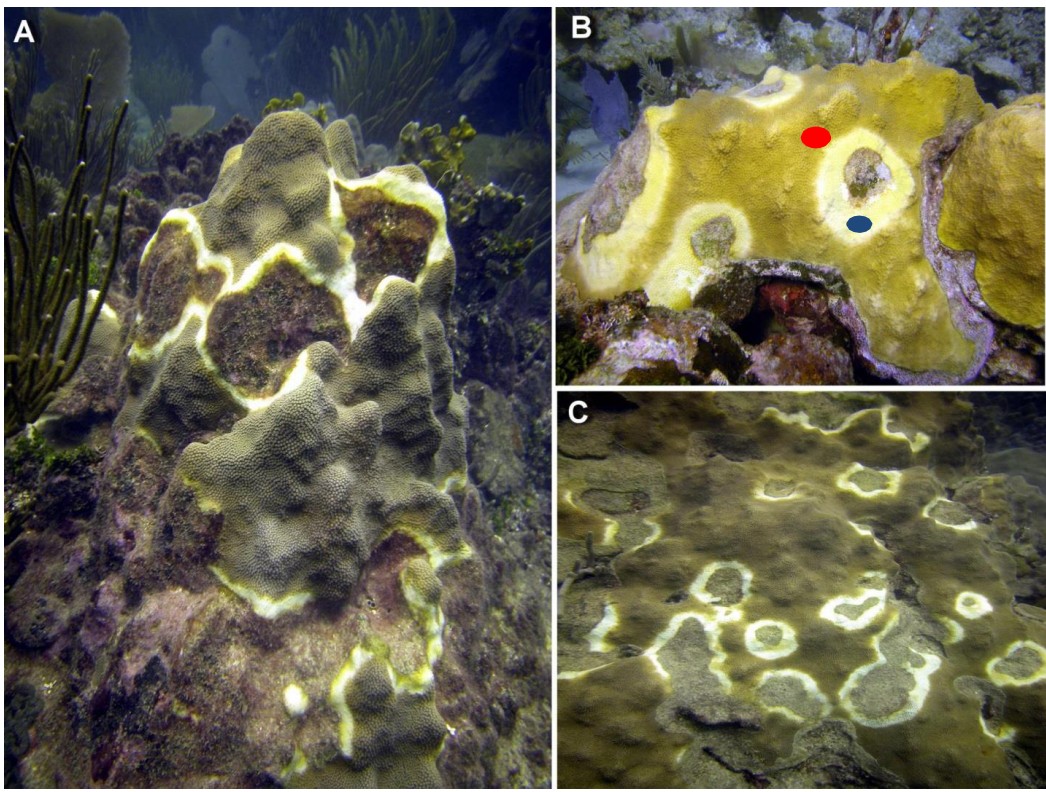

**Figure 1 Representative photographs of CYBD infected colonies.** (A) Several focal areas of initial stages of Caribbean Yellow Band Disease (CYBD in a large colony of *Orbicella faveolata* in La Parguera. (B) A large wide initial lesion of CYBD in *O. faveolata* with marks showing the sampled areas (red dot identifies the transition area and blue dot the CYBD active lesion). (C) Another large colony with advanced stages of the disease. (photographs E. Weil).

the ecological services they provide. The ability to detect and identify disease-affected tissue before visible signs of the disease are evident can be a useful diagnostic tool for monitoring efforts (*Anderson, Armstrong & Weil, 2013*).

Unfortunately, the development of diagnostic tools to predict and/or characterize disease progression is limited (*Pollock et al., 2011*). Representational Difference Analysis (RDA) is a form of subtractive hybridization that has been successfully used to detect stress responses at the level of transcription in cnidarians (*Morgan et al., 2012*). Developing gene expression biomarkers can be useful for monitoring health on coral reefs (*Kenkel et al., 2014*). The application of RDA to coral diseases represent a small scale approach to isolating critical transcriptional responses associated with healthy and/or diseased corals.

Sequencing of cnidarian genomes and transcriptomes has revealed a variety of potential pattern recognition receptors (PRRs) that could be used to detect both harmful and beneficial microbes and initiate signaling cascades including toll-like receptors, scavenger receptors, NOD-like receptors, and lectins (*Wood-Charlson et al., 2006; Miller et al., 2007; Shinzato et al., 2011; Meyer & Weis, 2012; Hamada et al., 2013; Poole & Weis, 2014; Ocampo et al., 2015*). In addition, cnidarians have many components of vertebrate innate immune pathways that PRRs may interact with to carry out cellular responses including

the complement system, Nf-$\kappa$B pathway, production of reactive oxygen species (ROS), antimicrobial peptides and the prophenoloxidase pathway (*Mydlarz et al., 2009*; *Shinzato et al., 2011*; *Vidal-Dupiol et al., 2011*; *Wolenski et al., 2011*; *Ocampo et al., 2015*). The involvement of these immune pathways in coral diseases is still not well understood. Several transcriptomic studies have identified differential expression of immune related transcripts between healthy and diseased states (*Closek et al., 2014*; *Libro & Kalaziak, 2013*; *Vidal-Dupiol et al., 2011*) One recent study on CYBD identified significant changes in the host transcriptome which included downregulation of two antimicrobial peptides and upregulation of two other immune genes in the diseased state as well as shifts in the microbial communities in tissues from healthy, diseased, and healthy border regions of the coral (*Closek et al., 2014*). From these data it is clear that each coral disease generates a unique response in the host and therefore characterization of differentially expressed genes between different stages of disease progression is essential for the development of diagnostic tools to predict and/or characterize the progression of a given disease. Characterizing specific responses in both members of the holobiont transcriptome are important steps in order to better characterize disease progression. The objective of this study was to identify responsive genes that can be used to better characterize the development of CYBD in the important reef-building coral *Orbicella faveolata*.

## MATERIALS AND METHODS

### Coral collections

Coral samples were collected in Oct 2009 and Oct 2013 from La Parguera, Puerto Rico. Specimens were collected under the General Collection Permit of the Department of Marine Sciences, University of Puerto Rico, Mayaguez. Diseased and healthy colonies of *Orbicella faveolata* were sampled from depth range of 9 to 12 m. At the time of collection, all samples were individually labeled, placed in plastic bags, and immediately transported back to the laboratory where they were placed on a seawater table and immediately processed. Five colonies with no visible evidence of disease were used as representative healthy tissue controls. Five other colonies with visible CYBD lesions were sampled as representative of the diseased condition. In addition, tissues were also sampled in the transition zone (*Weil, Cróquer & Urreiztieta, 2009b*), 2–4 cm in front of the visible lesion border on the same five diseased coral colonies. Diseased tissue (visible lesion) was identified and then separated from colony sample by chisel. Transition tissue was also taken from the same diseased colony. Transition tissues were sampled approximately 2–4 cm in front of the advancing visually identified transition/lesion border (see Fig. 1). Healthy colonies which had no visible signs of CYBD were also collected. After each sample was isolated, approximately 2 to 3 cm$^2$ of tissue was immersed in 5 mls of Trizol followed by immediate homogenization by vortexing. Total RNA was extracted following Trizol protocol (Invitrogen, Carlsbad, CA, USA) with the additional use of 2 ml phase-lock gels (5'Prime, Gaithersburg, MD) to aid in the recovery of the aqueous phase. RNA concentrations were estimated by absorbance readings at 260 nm (Biophotometer; Eppendorf, Hauppauge, NY). The integrity of the total RNA was confirmed by electrophoresis of an aliquot of

**Table 1 Primers used for RDA protocol.** RDA primer sequence information.

| Primer name | Sequence |
| --- | --- |
| R12 | 5′-GATCTGCGGTGA-3′ |
| R24 | 5′-AGCACTCTCCAGCCTCTCACCGCA-3′ |
| J12 | 5′-GATCTGTTCATG-3′ |
| J24 | 5′-ACCGACGTCGACTATCCATGAACA-3′ |
| N12 | 5′-GATCTTCCCTCG-3′ |
| N24 | 5′-AGGCAACTGTGCTATCCGAGGGAA-3′ |

each sample on a 1% formaldehyde agarose gel (*Sambrook, Firtsch & Maniatia , 2001*). Total RNA from each sample was purified by DNase I digestion followed by phenol/CHCl$_3$ extraction (Message Clean; GenHunter, Nashville, TN, USA). Messenger RNA (mRNA) was isolated (Oligotex; Qiagen, Valencia, CA, USA) from 100 µg of DNase I treated total RNA. Reverse transcription of 1 µg of mRNAs followed manufacturer's protocol (cDNA synthesis; Invitrogen, Carlsbad, CA, USA) which also included the use of both oligo-dT and random hexamer primers. The RT-reaction conditions were modified to 1 h at 37 °C, followed by 1 min at each temperature between 42 and 50 °C to maximize the number of full-length and partial transcripts copied (*Pastorian, Hawel & Byus, 2000*).

## Isolating RDA fragments

*Hubank & Schatz (1999)* provided the framework for the RDA protocol including RDA primer sequences (Table 1), with modifications to the amount of starting material (*Edman et al., 1997*) and the elimination of mung bean nuclease (*Pastorian, Hawel & Byus, 2000*). Double stranded cDNA from each treatment was digested with DpnII (New England Biolabs, Ipswich, MA, USA). Digested products were spin column purified to eliminate fragments smaller than 100 bp in length (QIAquick PCR purification; Qiagen, Valencia, CA, USA). Primers R12 and R24 were ligated onto the digested cDNAs at 15 °C overnight in 60 µl reactions. To generate sufficient quantities of the required amplicons necessary for downstream protocols, replicate PCR reactions were performed using 2.5 µl of ligated cDNAs that were amplified for 20 cycles. Prior to amplification, there was a 3 min incubation at 72 °C. Subsequently, primer was added and allowed to incubate for an additional 5 min. Once the entire 8 min heating had elapsed, Taq DNA polymerase was added and amplification proceeded for 20 cycles of 45 s at 95 °C followed by 3 min at 72 °C and then concluding for a final 10 min extension at 72 °C.

Gel electrophoresis confirmed the size distribution within each amplicon. Twenty replicate PCR reactions were pooled and then precipitated by isopropanol to concentrate. Each amplicon was quantified by UV spectrophotometry and subsequently diluted with TE buffer to a final concentration of 0.5 µg/µl. Amplicons that were digested with DpnII (New England Biolabs, Ipswich, MA, USA) to remove R24 primers became the cut-drivers which were used in downstream reactions. Two rounds of hybridization were performed in this investigation. With the assistance of J12 primer, the J24 primer was ligated onto cut-driver and used as the tester amplicon in the first round of hybridization. Tester

populations in the second round of hybridizations used the N24/N12 primers. Ligation conditions were always the same throughout the investigation even though the use of J24/J12 or N24/N12 primers depended on the round of hybridization. The ratio of tester/driver in round one was 1:100, whereas in round 2 it was 1:800. The first round of hybridization combined 50 ng of J24-ligated tester to 5 μg of cut-driver. The second round of subtraction/hybridization combined 6.25 ng of N24-ligated hybridization product from the first round of hybridization and 5 μg of cut-driver. Each sample population (healthy or diseased) was used as a tester in one series of hybridizations and as the driver in the other series of hybridizations. After combining testers and drivers in the desired ratio, pooled samples were extracted with Phenol/Chloroform/Isoamyl alcohol and then precipitated in 30 μl of 10 M ammonium acetate and 250 μl ethanol at −70 °C for one hour. After centrifugation at 14,000 rpm for 15 min at 4 °C, resulting pellets were washed twice in 70% ethanol and allowed to air dry. Each pellet was resuspended in 4 μl EEx3 (30 mM EPPS, pH 8.0 at 20 °C, 3 mM EDTA) buffer by pipetting repeatedly for 2 min then warmed to 37 °C for 5 min, vortexed, and then briefly centrifuged. Samples were then overlaid with 35 μl mineral oil and then heated to 95 °C for 5 min to denature. Afterwards they were allowed to cool to 67 °C and 1 μl 5M NaCl was added directly into the DNA, and the samples were incubated overnight at 67 °C. After the overnight incubation, mineral oil was removed and samples were diluted in 95 μl TE. Five microliters of a hybridization product were used in downstream PCR reactions. The hybridization products were amplified by PCR beginning with a 3 min incubation at 72 °C followed by the addition of Taq DNA polymerase. After 5 additional minutes at 72 °C, primer was added (either J24 or N24 primer depending on which round of hybridization had been performed). Amplification conditions consisted of 45 s at 95 °C followed by 3 min at 70 °C (J24) or 72 °C (N24) for a total of 27 cycles and then concluded with a 10 min extension at 72 °C. To eliminate the need of mung bean nuclease, a modified PCR reaction was employed which involved taking a 10 μl aliquot out of the PCR reaction after 7 cycles and placing it in a new PCR reaction with all reagents and continuing for an additional 20 cycles (*Pastorian, Hawel & Byus, 2000*). Amplified RDA products were cloned (TopoTA, Invitrogen, Carlsbad, CA, USA) and then sequenced using M13 (forward and reverse) primers (Nevada Genomics Center, University of Reno, NV, USA). Ninety-five sequences from CYBD infected tissues were cloned and sequenced. Another 190 sequences were sequenced from RDA products representing healthy tissues.

## Sequence analysis

A contig assembly program (CAP3) (http://doua.prabi.fr/software/cap3) was used to determine how many unique sequences were represented within the total number of cloned RDA products. Sequences were compared against the nr and EST databases at NCBI (National Center for Biotechnology Information) database (http://www.ncbi.nlm.nih.gov) using the BLASTX algorithm with default parameters. CAP3 analysis indicated there were 9 contigs and 53 singletons.

**Table 2  RDA candidates screened CYBD analysis.** Fourteen RDA products screened for inclusion in qPCR assays. Probes 1–5 were screened as potential qPCR control genes. Probes 6–14 were screened as potentially differentially expressed transcripts.

|    | RDA probes | Putative genes of interest |
|----|------------|----------------------------|
| 1  | H40D1-30   | Ribosomal s8               |
| 2  | H40D1-50A  | 18s rRNA gene              |
| 3  | H4D1-22    | Poly-A binding protein     |
| 4  | H3D1-20    | Ribosomal protein          |
| 5  | H3D1-30    | Ribosomal protein L27a     |
| 6  | H40D1-46   | Tetratricopeptide repeat   |
| 7  | H4D1-20    | Predicted PHD-finger       |
| 8  | H3D1-4     | Ubiquitin ligase           |
| 9  | H4D1-1     | DNA J like                 |
| 10 | H4D1-2     | Cytochrome oxidase subunit 1 |
| 11 | H40D1-55   | Superoxide dismutase-like  |
| 12 | H40D1-56   | skp1 family protein        |
| 13 | H40D1-60   | IG domain                  |
| 14 | H40D1-64   | Early growth response      |

## Virtual northern dot blot screen

Two microliter aliquots of individually amplified RDA products were blotted onto replicate positively charged nylon membranes. Amplicons were amplified by PCR using a DIG-labeled nucleotide (Roche, Indianapolis, IN, USA). The presence/abundance of individual RDA products was detected by chemiluminescence and quantified by densitometry (*Morgan et al., 2012*). One of the replicate membranes of RDA products was probed with DIG-labeled amplicon from the healthy tissues, while the other replicate membrane was probed with DIG-labeled amplicon from the diseased tissues. DIG-labeled RDA products with differences in signal intensities between the identical membranes were identified and selected as candidate GOIs for subsequent qPCR analyses.

## Selection of GOIs and primer creation

Virtual Northerns (*Franz, Bruchhaus & Roeder, 1999*) and BLAST analyses identified 14 potential genes of interest (GOIs) that were evaluated as candidates as a qPCR control gene or a gene which is differentially expressed amongst the tissues (Table 2). Gene specific forward and reverse primers were created for the GOIs using Primer3 (http://frodo. wi.mit.edu/primer3/) (Table 3). Primer efficiency was initially screened by amplifying a target cDNA over a range of $T_m$'s to determine optimal annealing (Mastercycler gradient; Eppendorf, Hauppauge, NY) and then compared predicted amplicon size to observed amplicon size on a 2% TBE agarose gel. Five GOIs were chosen for further analyses (Table 4). Poly-A binding protein (PABP) was chosen as the reference gene because it has previously demonstrated a consist pattern of expression as a qPCR reference gene (*Rodriquiez-Lanetty et al., 2008*). Four of the five genes (EGR, Ubiquitin Ligase, Superoxide dismutase, and PABP) were representative of the coral host while the fifth

**Table 3  Primers for GOIs.** Genes of Interest and their corresponding primers used in qPCR reactions.

| RDA Probe ID | Putative Gene homolog | Primers | Amplicon length (bases) | Annealing temp |
|---|---|---|---|---|
| H40D1-64 | Early growth response | F: TGAACAGATTTGCGACGTTT<br>R: AGCCCCCAACTGTCTCTCTT | 154 | 56 °C |
| H4D1-22 | Poly-A binding protein | F: TCGGTGTCAAAATGGACAAA<br>R: ATCCTTCCCTTCGCAAATCT | 178 | 54 °C or 56 °C |
| H4D1-2 | Cytochrome oxidase | F: TGGAAAGGATGGGATTCTTC<br>R: TGAATGGAGAAAAGATTGTTGC | 164 | 56 °C |
| H3D1-4 | Ubiquitin ligase | F: GGCATTTTAACGGGGTCTTT<br>R: GTTGGGTGATGAGACGGACT | 165 | 54 °C |
| H40D1-55 | Superoxide dismutase | F: CAGGAACTGGAACCGATGAT<br>R: TTACCGACGTCGACTATCCA | 168 | 56 °C |

**Table 4  BLAST results for RDA probes.** Searches were performed using BLASTX at NCBI using the non-redundant database (nr) with default search parameters.

| RDA probe | Accession | E-value | Putative homolog | Organism ID | Homolog Accession # |
|---|---|---|---|---|---|
| H4D1-22 | JZ875039 | $2e^{-54}$ | Poly-A binding protein | *Nematostella vectensis* | XP_001625306 |
| H40D1-64 | JZ875040 | $1e^{-14}$ | Early growth response | *Echinococcus granulosus* | EUB53836 |
| H4D1-2 | KT149212 | $6e^{-49}$ | Cytochrome oxidase | *Symbiodinium microadriaticum* | ABK57976.1 |
| H3D1-4 | JZ875037 | $2e^{-14}$ | Ubiquitin ligase | *Rattus norvegicus* | XP_221191.5 |
| H40D1-55 | JZ875038 | $1e^{-10}$ | Superoxide dismustase-like | *Saccoglossus kowalevskii* | XP_002734284 |

gene (Cytochrome Oxidase subunit 1) was representative of zooxanthellae. Melt-curve analysis was performed to determine specificity of priming. Individual amplicons were also extracted from agarose gels (Qiaquick gel extraction kit, Qiagen, Valencia, CA, USA) and sequenced to confirm amplification of the intended target.

## Quantitative real-time PCR

Quantitative Real Time PCR (qPCR) assays were performed in four replicate 25 µL reactions. The components within each reaction were: 12.5 µL (2X) *Power*SYBR® Green (Applied Biosystems, Carlsbad, CA), 2.5 µL of forward and reverse primer (10 µM each), 2.5 µL of 3-fold diluted cDNA from reverse-transcription reaction, 0.25 µL Taq DNA polymerase (BioLabs, Ipswich MA), and 4.75 µL DI water. All qPCR reactions were performed on a StepOne™ Real Time PCR machine (Applied Biosystems, Carlsbad, CA). Reaction conditions involved heating at 95 °C for 15 s, annealing for 54 °C or 56 °C for 20 s, and then elongating at 72 °C for 20 s. All reactions were monitored by a melt curve analysis to ensure specific amplification and absence of primer dimerization. Reactions were ramped from 60 °C to 95 °C at a rate of 0.3 °C s$^{-1}$.

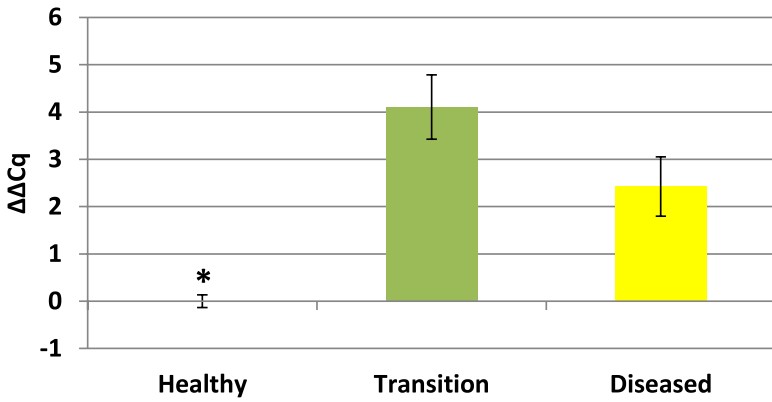

**Cytochrome Oxidase**

**Figure 2 Expression of zooxanthellae Cytochrome Oxidase.** The $\Delta\Delta$Cq values represent transformed expression of cytochrome oxidase relative to PABP expression. An * represents a condition that was significantly different in expression ($P < 0.01$) relative to other conditions. Error bars represent $\pm$ SE.

## Expression analysis

Replicate Cq values for each GOI were averaged to determine $\Delta$Cq and $\Delta\Delta$Cq for each sample within a health condition. All qPCR $\Delta$Cq and $\Delta\Delta$Cq values are based on the consistent expression of the PABP across all samples in this study. The $\Delta\Delta$Cq method was used to determine the differences between targeted GOIs and a single reference gene (*Bustin et al., 2009*). One-way ANOVA was used on the $\Delta\Delta$Cq data to identify significant differences in the expression of an individual GOI within tissues representing different stages of the disease (i.e., diseased, transition, healthy). Similarities in variance were determined by Levene's Test of Equality of Error Variances. If the variance between disease stages was similar, then the Student-Neuman-Keuls (SNK) posthoc test was performed to determine which group(s) were significantly different from the rest. If variance between disease stages was different, then Tamhane's T2 posthoc test was applied since One-way ANOVA is generally insensitive to heterosedcasticity.

## RESULTS

Cytochrome oxidase had its highest average expression in transition zone tissues (Fig. 2). One-way ANOVA indicated that cytochrome oxidase expression was significantly lower ($P < 0.01$) in healthy tissues compared to transition and diseased tissues (see Fig. 2). The highest average expression for the EGR was in the diseased tissues and there were significant differences in the levels of expression ($P < 0.01$) of EGR in healthy tissues compared to transition and diseased tissues (Fig. 3). Ubiquitin ligase had its highest average expression in the diseased tissues and there were significantly differences in expression ($P < 0.01$) in diseased tissues compared to transition and healthy tissues (Fig. 4). Expression of superoxide dismutase-like (SOD) was found to be similar across all samples, therefore no significant differences in expression (data not shown).

Analysis was also performed on the collective expression of all three genes to determine whether any significant differences existed between health conditions. One-way ANOVA of

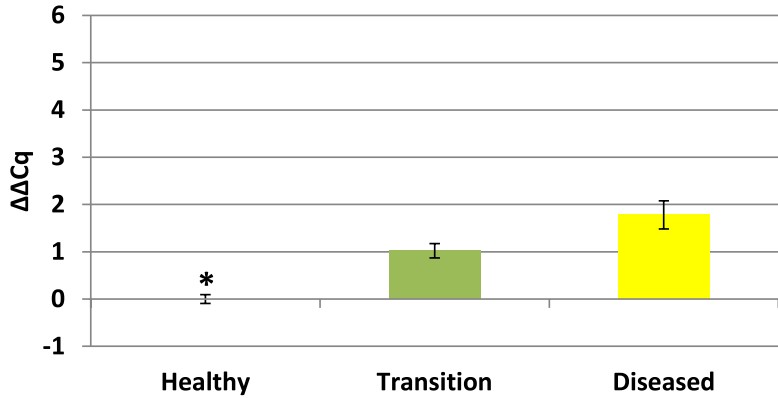

**Figure 3 Expression of early growth response (EGR).** The $\Delta\Delta$Cq values represent transformed expression of the EGR relative to PABP expression. An * represents a condition that was significantly different in expression ($P < 0.01$) relative to other conditions. Error bars represent $\pm$ SE.

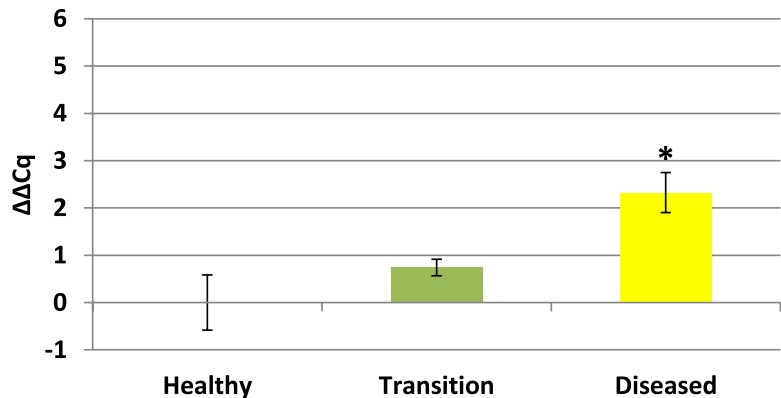

**Figure 4 Ubiquitin Ligase expression.** The $\Delta\Delta$Cq values represent transformed expression of ubiquitin ligase relative to PABP expression. An * represents a condition that was significantly different in expression ($P < 0.01$) relative to other conditions. Error bars represent $\pm$ SE.

health conditions also reaffirmed that there is a significant difference ($p < 0.001$) between the healthy condition and the two different conditions associated with diseased colonies.

## DISCUSSION

The purpose of this study was to characterize the expression profiles of a small suite of GOIs in the Caribbean coral *Orbicella faveolata* exhibiting CYBD. This study does not identify the pathogen or an associated virulence factor, but does identify transcriptional responses uniquely associated with the coral host as well as the algal symbiont. *Pollock et al. (2011)* states the validity of a diagnostic test is determined by its ability to distinguish host organisms that have the disease from those that do not. This study demonstrates the application of a small suite of genes that can differentiate between healthy colonies (no

disease), diseased colonies (visible lesions), and tissues that are nearby (transition area) and precede visible lesions on diseased colonies. These three GOIs provide quantitative measurements of location-dependent changes in their levels of transcription. Expression patterns of these GOIs expand our understanding of the cellular processes associated with the development/progression of CYBD. This information also advances our understanding of how the disease impacts tissues that precedes visible lesions. The data herein also provides additional supporting evidence that CYBD is systemic as proposed by *Weil, Cróquer & Urreiztieta (2009b)*, affecting tissues that do not show signs of the disease in infected colonies.

## Functional significance of GOI's

### Cytochrome oxidase

Elevated expression of cytochrome oxidase in diseased and transition tissues reveals an increased demand for energy. Diseased tissues have previously been characterized as having significantly fewer *Symbiodinium* (*Cervino et al., 2004a*; *Cervino et al., 2008*), and yet another study indicates that zooxanthellae densities are not necessarily lower in diseased tissues (*Mydlarz et al., 2009*). Concurrently, transition tissues visually appear asymptomatic (*Weil, Cróquer & Urreiztieta, 2009b*; *Anderson, Armstrong & Weil, 2013*; *Closek et al., 2014*). By comparison, healthy colonies with normal zooxanthellae densities do not exhibit elevated expression of cytochrome oxidase (see Fig. 2). Zooxanthellae densities were not quantified in this study. Cytochrome oxidase expression reveals that mitochondrial function in zooxanthellae is impacted by CYBD. Growth involves higher energy demands and investments which can have significant impacts on the corals ability to respond to infectious disease and other stressed conditions such as bleaching (*Pinzon et al., 2014*). Mitochondria as well as chloroplasts generate ATP for biosynthetic reactions in autotrophic organisms. Both organelles are metabolically linked through the $C_2$ pathway (*Bauwe, Hagemann & Fernie, 2010*). Little attention has been given to the interaction of mitochondria and chloroplast in stressed *Symbiodinium*. As the balance between photosynthesis and photorespiration fluctuates in favor of photorespiration in chloroplasts, photorespiration facilitates energy production in mitochondria (*Tcherkez et al., 2008*; *Bauwe, Hagemann & Fernie, 2010*) . In CYBD tissues, chloroplasts are intact but grana and thylakoid membranes appear disorganized (*Cervino et al., 2004a*). In addition, there is evidence that photosynthetic pigments exhibit changes in spectral features within CYBD tissues (*Anderson, Armstrong & Weil, 2013*). Such changes are consistent with diminished light-capturing capabilities of the photosynthetic machinery which would tilt the balance of photosynthesis/photorespiration in favor of photorespiration. Up-regulation of cytochrome oxidase in zooxanthellae from colonies exhibiting CYBD provides new molecular information that is consistent the conclusions of *Cervino et al. (2004a)*, *Cervino et al. (2008)* that CYBD is a disease of *Symbiodinium*. The elevated expression of cytochrome oxidase offers molecular evidence that zooxanthellae are exhibiting increased demands for energy. This zooxanthellae response for increased energy production is distinctively different from the lack of investment in energy production

by the coral host as identified by *Closek et al. (2014)*. The cytochrome oxidase coupled with findings from *Closek et al. (2014)* reaffirms the conclusions of *Pinzon et al. (2014)* that energy demands are intimately linked to holobiont's ability to responds to stressed conditions. Quantifying expression of an algal gene (i.e., cytochrome oxidase) relative to PABP expression of the coral host is a valid comparison because both genes are representative of the transcriptional activity occurring in the holobiont at the time of tissue sampling.

### Early growth response

Preliminary BLASTX results suggest that RDA product H1D1-64 is an early growth response (EGR) which is known as a transcription factor (*Barshis et al., 2013*). This gene is known to be associated with the initiation of immune responses (*McMahon & Monroe, 1996*), mitogenesis and cell growth (*Sukhatme, 1990*), and tumor suppression (*Huang et al., 1997*). Corals are known to have a complex repertoire of immune responses for how corals respond to a pathogen (*Sutherland, Porter & Torres, 2004*; *Miller et al., 2007*; *Schwarz et al., 2008*; *Kvennefors et al., 2010*; *Shinzato et al., 2011*; *Poole & Weis, 2014*). Coral immune mechanisms include coral wound healing, hemocytosis, phagocytosis, encapsulation, and basic immunological memory to fight off pathogens (*Sutherland, Porter & Torres, 2004*; *Palmer & Traylor-Knowles, 2012*). EGR has also previously demonstrated significant expression in corals responding to heat stress (*Barshis et al., 2013*) which is a well-recognized stressor that can induce bleaching. There is a growing body of evidence of linkages between bleaching and immune responses in cnidarians (*Weis, 2008*; *Kvennefors et al., 2010*; *Detournay et al., 2012*; *Pinzon et al., 2014*; *Pratte & Richardson, 2014*). Elevated EGR expression in the transition tissues (see Fig. 3) may represent an important signal associated with a preliminary stage of infection in nearby tissues, but further studies will need to be conducted to clarify the role of this protein in the progression of CYBD.

### Ubiquitin ligase

Ubiquitin ligase in this study is representative of RING-type E3 ligases. Ubiquitination directs many cellular functions including protein degradation (*Komander & Rape, 2012*) and regulating a variety of cellular processes including vesicle trafficking (*Hsu et al., 2014*), cell cycle control (*Skaar, Pagan & Pagano, 2013*), and immune responses (*Jiang & Chen, 2012*). Results herein reveal ubiquitin ligase is initially up-regulated in transition tissues, but peaks with highest expression in visible lesion tissues (see Fig. 4). Elevated expression of ubiquitin ligase in diseased tissues suggests this profile is representative of a later stage in the development of the disease. Elevated expression of ubiquitin ligase in diseased tissues is particularly interesting for a couple of reasons. Microbes are known to highjack host ubiquitin pathways in order to manipulate host signaling to facilitate bacterial infection and proliferation (*Zhou & Zhu, 2015*), and *Closek et al. (2014)* identified CYBD diseased tissues having 2-3 times greater bacterial diversity compared to healthy tissues while the transition tissues actually had the highest species richness. Ubiquitination is also known to be associated with lysosomal degradation of plasma membrane proteins (*Komander & Rape, 2012*), therefore the expression profile of ubiquitin ligase in this study

is also consistent with elevated expression of lysosomal-like enzymes and anti-microbial responses identified by *Mydlarz et al. (2009)*. Ubiquitin ligase expression in this study may represent the nexus between the microbial diversity identified by *Closek et al. (2014)* and the anti-microbial responses identified by *Mydlarz et al. (2009)*.

**Expression profiles enhance findings of previous CYBD studies**

Results from this study show that transition tissue 2–4 cm away from a visible lesion exhibits a different expression profile compared to visible lesion tissues (see Figs. 2–4). These results reaffirm conclusions of previous studies (*Mydlarz et al., 2009*; *Weil, Cróquer & Urreiztieta, 2009b*; *Anderson, Armstrong & Weil, 2013*; *Closek et al., 2014*) that characteristic differences exists between healthy and diseased colonies, as well as differences between diseased tissues and asymptomatic tissues that precede visible lesions on diseased colonies.

Results in this study coupled with *Closek et al. (2014)* findings that microbial diversity was highest in tissues the precede visible lesions raise the intriguing possibility that pathogen(s) responsible for CYBD may reside within the microbial community of the transition tissues. If the etiological agent for inducing CYBD actually resides in the microbial community of the transition tissues, then disease progression through a colony may actually follow a pattern similar to secondary succession observed in terrestrial communities after a disturbance such as a fire. The visible demarcation between transition tissues and lesion tissues corresponds to "fire line" of the disturbance. Consequently, lesion tissues behind the "fire line" are committed to subsequent degradation. Therefore the *Symbiodinium* response in transition tissues represents a potential sentinel of the approaching disturbance.

*Mydlarz et al. (2009)* investigated the anti-oxidant responses of prophenoloxidase (PPO) and peroxidase (POX) in CYBD tissues. Without the presence of significant ROS, down-regulation of PPO and POX in CYBD tissues would be expected when amounts of reactive oxygen species (ROS) generated is minimal. As previously identified, there was no significant up-regulated expression of SOD in coral host tissues (data not shown) which is congruent with the down-regulation of the antioxidant enzymes quantified by *Mydlarz et al. (2009)* and the absence of SOD expression identified by *Closek et al. (2014)*. Concurrently, the mitochondrial response of elevated cytochrome oxidase further suggests a potential link between the absence of detectable ROS and the metabolic process of photorespiration which integrates the functions of mitochondria, chloroplasts, and peroxisomes through the photorespiratory $C_2$ cycle (*Bauwe, Hagemann & Fernie, 2010*; *Voss et al., 2013*).

## CONCLUSIONS

This study has characterized CYBD in the Caribbean coral *Orbicella faveolata* by quantifying the expression patterns of three GOIs (cytochrome oxidase, EGR, and ubiquitin ligase). Individually, expression of these GOIs reveal altered physiology that can be attributed to a wide variety of stressed conditions. Collectively, these GOIs represent a potential diagnostic tool capable of differentiating between healthy and diseased colonies as well as between two different stages of CYBD within a diseased colony. The expression profiles of

these three GOIs in transition tissues compared to healthy and diseased tissues suggest the initial development of this disease begins before visible lesions are evident. Quantification of a small suite of genes by qPCR from within a complex pool of animal and algal transcripts reaffirms the specificity of the technique previously demonstrated by *Seneca et al. (2010)* and achieves the goal of sensitivity and specificity of a molecular diagnostic tool to understand the cellular events associated with disease pathogenesis as outlined by *Pollock et al. (2011)*. The collective expression of these three GOIs produces expression profiles that are uniquely different between transition, diseased, and healthy tissues. Collectively, expression of these GOIs provides greater resolution for differences that exist between tissues on both sides of the well-defined reference point of the transition/lesion border of CYBD colonies. Biomarkers are most effective when they can detect stress signals that vary in intensity (*Tyler & Gretchen, 2012*; *Kenkel et al., 2014*). Signal intensity increases (healthy < transition < diseased) for EGR and ubiquitin ligase, but signal intensity peaks (healthy < transition > diseased) for cytochrome oxidase. While the expression profile of each GOI is different from the other GOIs, it is their collective expression can discern differences that the physiological status of healthy, transitional, and diseased tissues. The diagnostic power of these GOIs remain tentative without more extensive sampling to account for genotypic variation of individual colonies and the distance tissues are sampled from the visible demarcation of a lesion. Future transect sampling can provide greater resolution about the distance from a visible lesion that these signals can be detected which may help to further characterize the size of a "virulence wave" that precedes visible lesions. The data in this study expands the current understanding of the molecular responses associated with CYBD infections and the results identify significant responses associated with different members of the holobiont. Lastly, this suite of GOIs offers an assessment tool that may provide greater resolution to the spatial distribution of colonies susceptible to infections of CYBD on a reef. Studies such as *Soto-Santiago & Weil (2012)* which quantified the spatial distribution of CYBD at various reef locations may find greater resolution in the distribution patterns of CYBD by incorporating these GOIs to screen for colonies in early stages of this disease that precede visible lesions. Detection of colonies in the earliest stages of CYBD may also help to spatially identify colonies at greatest risk of developing CYBD.

As our understanding of how environmental and anthropogenic stressors influence the transmission of CYBD continues to expand, there will be increasing demands for rapid assays capable providing resource managers with relevant field monitoring information to make timely decisions. Future studies may also use these GOIs to assist in further characterizing other stressed conditions such as bleaching. The methodologies employed herein can be applied to studies of other coral diseases as well. Each coral disease with a unique etiology will have its own distinctive transcriptional "finger print" representing critical metabolic pathways that are impacted on a specific temporal/spatial scale. While whole transcriptome analysis maybe ideal, many labs are not financially equipped to employ such analyses. RDA represents an inexpensive alternative to transcriptome-wide assays used to identify critical components of a specific stressed condition. Both RDA and/or transcriptome analysis can be used to identify GOIs of critical biochemical

pathways for CYBD or any other coral disease. Once candidate GOIs are identified, they can be coupled with qPCR and used to generate a transcriptional profile uniquely associated with a particular coral disease. Such assays are capable of providing valuable transcriptional information within hours of tissue sampling.

## ACKNOWLEDGEMENTS

We wish to acknowledge the laboratory assistance of Peggy Molyneux and Preston Neely. The Department of Marine Sciences, University of Puerto Rico Mayaguez provided logistical support and lab space.

### Funding

Financial support was provided by Berry College for the Richards Scholarships to Elizabeth Stepp and Cydney Wilbanks; an NSF-REU (project # 0354017) awarded to Berry College which funded Angela Poole; and the Berry College Provost's Office for travel monies to Kylia Goodner. Additional funding was provided by NSF IOS # 1017510 and OCE-1105143 to EW. The funders had no role in study design, data collection and analysis, decision to publish, or preparation of the manuscript.

### Grant Disclosures

The following grant information was disclosed by the authors:
Berry College for the Richards Scholarships.
NSF-REU: # 0354017.
The Berry College Provost's Office.
NSF IOS: # 1017510, OCE-1105143.

### Competing Interests

The authors declare there are no competing interests.

### Author Contributions

- Michael Morgan conceived and designed the experiments, performed the experiments, analyzed the data, contributed reagents/materials/analysis tools, wrote the paper, reviewed drafts of the paper, logistical support and lab space.
- Kylia Goodner and James Ross performed the experiments, analyzed the data, wrote the paper, prepared figures and/or tables, reviewed drafts of the paper.
- Angela Z. Poole performed the experiments, analyzed the data, wrote the paper, reviewed drafts of the paper.
- Elizabeth Stepp performed the experiments.
- Christopher H. Stuart performed the experiments, analyzed the data.
- Cydney Wilbanks performed the experiments, analyzed the data, prepared figures and/or tables.

- Ernesto Weil analyzed the data, contributed reagents/materials/analysis tools, wrote the paper, reviewed drafts of the paper, logistical support and lab space.

## Field Study Permissions

The following information was supplied relating to field study approvals (i.e., approving body and any reference numbers):

Specimens were collected under the General Collection Permit of the Department of Marine Sciences, University of Puerto Rico, Mayaguez.

## DNA Deposition

The following information was supplied regarding the deposition of DNA sequences:

GenBank accession numbers: JZ875039, JZ875040, KT149212, JZ875037, JZ875038

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
