# Peer review of "Development and application of molecular biomarkers for characterizing Caribbean Yellow Band Disease in Orbicella faveolata"

_PeerJ, doi:10.7717/peerj.1371_

## Round 0.1 · original submission · Major Revisions

· Academic Editor

Major Revisions

All reviewers are enthusiastic about the results but raise valid points that will make this a more valuable contribution. Please restructure the manuscript so there is more clarity about the RDA method in the introduction. Tone down the statement about the reliability of the diagnostic tool. Tighten the different sections so the manuscript is more concise. Try to include all the other minor editorial suggestions that will also improve the readability of your paper.

Reviewer 1 ·

Basic reporting

This manuscript uses RDA analysis to determine diagnostic tools for detection of disease. The data are interesting, although there is not much discussion of the RDA technique and why it is appropriate for this study when setting up the experiment. I found this information in the conclusions, but it should be moved up. While the changes in expression between healthy and diseased colonies are interesting, the ability to use these 3 genes in a diagnostic framework are bit overstated.

Experimental design

The introduction is missing some key information. There is no discussion of the techniques such as RDA that may be able to solve some of the issues the authors bring up in the introduction. Also there are references missing that could explain some of the issues with CYBD. There are several papers that examine immunity in O. faveolata (both protein, antimicrobial activity and gene expression), but are not included in a discussion of immunity on line 105. There is also very little discussion about what we know about coral immunity in general terms, especially since this paper seeks to find biomarkers for the disease, a discussion of the known key responses to it is critical.

The results include a lot of methods that distract from the actual data. Nearly all of the results verbiage should be moved into methods.

Figure 5 is repetitive, and either not necessary or figures 2-4 aren’t necessary. I actually like all the genes together on 1 graph. The graphs need polishing, there are spelling errors and the colors could be improved.

The discussion is really difficult to follow and not structured well to highlight the results. Its really, really long and can be cut down and streamlined in many places.

Line 309 – but see Mydlarz et al. 2009

Line 307 – how do these data add support to the conclusion that CYBD is a disease of the Symbiodinium? This doesn’t seem to be a point of debate as its clear that isolated zooxanthellae are definitely diseased and appear markedly different from those in the transitional zone or a healthy colony. Did the authors in this current study count zooxanthellae, because as mentioned above, there aren’t always fewer cells in the CYBD affected areas. There also is no discussion of how the upregulation of cytochrome oxidase helps the readers understand more about what functionally may be going on in the diseased areas. Cytochrome oxidase is a protein in the mitochondria of eukaryotes involved in respiration, how does the function of this gene help us understand the differences between diseased and healthy tissue? It would seem that the symbionts are stressing out and respiration is going through the roof, probably because the bacteria are present. BUT this type of response could be due to almost ANY stressor.

Line 340 and also line 352- why is this result in the middle of the discussion?

Line 379 – already mentioned this above. A 1 page long discussion of the transitional zone is not necessary. This study does reaffirm some previous results, which is fine, just keep it short.

Line 390- I am unsure if this paper really supports the use of these genes as biomarkers. It seems they were chosen because they had different expression between healthy and diseased colonies, which IS interesting. But there was not much thought to choosing biomarkers that make sense to this disease. Therefore I have a lot of trouble with the spin of this manuscript. Like I mentioned earlier, this data is good. Its interesting to know what genes are significantly different between disease and healthy colonies to understand the process of infection and immunity/response, but it just doesn’t make a lot of sense in this ‘diagnostic’ framework. Especially since this section dedicated to the ‘development of a diagnostic tool’ does not present any more detailed information about how this would be accomplished. It just repeats and reiterates the data found. Also, how do we actually know if the early signs of disease would lead to an increase in the same genes before signs of yellowing or active infection? The disease section of the infected colony is the end-point of the disease before its going to die, and as the authors mention themselves, the transitional zones are variable in the response to these genes and even previous work showed variability of the transitional zones in expression of genes, proteins, etc. depending on how far they are from the active infection. I am just not sure these connections can be made at this time.

Line 426 – the point by point discussion of these findings compared to Closek are completely unnecessary. These points can be discussed within each discussion of the genes. Its just goes on too long and does not present any more information.

Line 463 – same issue with these last conclusions, they go on way too long. Ultimately this paper looked at 3 genes, I am unsure why the discussion is 11 pages long. Some of the missing information on the function of the genes is in these conclusions, but absent from the specific section on that gene? Its very confusing and needs some tightening up. Much of it can be deleted as it is repetitive, or moved to places where it is needed like the introduction (lines 506-510) on the rationale of using RDA, for example.

Validity of the findings

The findings are indeed valid, just not in the 'biomarker' and 'diagnostic' arena that the authors are stating.

Additional comments

Ultimately this study has legs, it can be a good contribution to the gene expression changes in diseased tissues. However, this manuscript needs a lot of revisions, reorganization, tightening and shortening to make it ready for publication.

Reviewer 2 ·

Basic reporting

Table 4 should include information on the SOD-like gene.

I don’t think that Fig 5 adds more information to the manuscript.
According to the authors criteria, Fig 5 indicates that both cnidarian genes showed differential expression between control and the transition/diseased conditions, so their claim is apparently well supported. However, ubiquitin was not different between control and the transition condition (Fig 4) which would make Fig 5 controversial. Fig 5 also presents the same results as Figs 2, 3 and 4, but analysed differently, substituting statistical information on comparisons within each gene for an overall comparison among conditions, which I consider unnecessary.

It is not specified if raw data for Ct values of all GOIs have been submitted using RDML.

The discussion needs to be more concise. I would suggest not to break the functional significance of isolated GOIs into the three genes investigated, and incorporate the observation of distance within the discussion rather than make a separate section for it.
Subheadings for the discussion should be less: Functional significance of GOIs, and Development of a diagnostic tool, should be sufficient.

Citations in the discussion section need correction. Many multiple author references were cited by the first author only, followed by the date. For example Closek 2014, Weil 2009, etc.

Experimental design

Information for coral samples (how many per conditions, how much tissue or what size of nubbins, etc.) is given in the results section rather than in the methods section. I would suggest to correct this.

Even though transcript levels were high for cytochrome oxidase, and with lower symbiont numbers transcript levels could have been underestimated, the authors did not determine symbiont density in any of the coral samples, thus "lower symbiont numbers" is nevertheless speculative, and this needs to be stated as such.

The experimental quantification of gene expression by qRT-PCR used only a single gene as reference for coral tissues, and none for symbionts.

I am afraid that to include reference genes for symbiont transcript level quantification, the experiments will need to be repeated. However, to include an additional reference gene for coral tissues, perhaps there is some gene in the candidate RDA products that might be useful.

As to qPCR MIQE standards, it is not specified which method was followed for the calculation of Cq values.

Validity of the findings

The authors discuss that assessing gene expression evaluated in reference to a single gene for the holobiont is valid, because both genes (PABP and any algal gene) are representative of the transcriptional activity occurring in the holobiont. But it is clear that two different organisms are involved, and one cannot overlook the fact that gene expression follows different cues and responses in each organism. Also Symbiodinium is not a typical organism regarding gene regulation and expression (see for example, Bayer et al. 2012 [PLoS ONE 2012, 7:e35269]; Lin et al. 2010 [Proc Natl Acad Sci 2010, 107:20033–20038]). The importance in having two reference genes for each symbiotic partner relies in the quality of information one can acquire from studying transcript levels. For instance, if high temperature preceded the disease, cytochrome oxidase may have been upregulated due to temperature, and thus may not be directly implicated in the disease. Then finding corals with increased expression for this gene would wrongly identify them as diseased.

Additional comments

In my opinion this contribution has great potential for the identification of genes that can be used as a tool for CYBD early detection, and also to uncover the mechanisms of CYBD in corals. But the results as presented are rather preliminary; the validation of the RDAs by RT-PCR is incomplete.

Reviewer 3 ·

Basic reporting

No Comments

Experimental design

No Comments

Validity of the findings

Transcription occurs quickly. As such, based on the sampling methods provided and that the samples were not flash frozen, I recommend that the authors address the possibility that sample differences could be due to handling/processing (e.g. differences in length of time after sampled from the colony until it was placed in trizol) or argue otherwise and provide individual sample profiles in supplemental figure(s).

Additional comments

Coral disease studies are severely needed and the authors’ effort to investigate CYBD is applauded. I would recommend the article for publication given that a few minor items listed below are addressed. (line number(s): followed by comment and/or suggestion)

68-72: Please cite the works that show mortality in Dendrogyra and Pseudodiploria caused by CYBD.
242: “significantly different” should be changed to “different”, as something is either significant or not
285: pluralize "gene"
312-314: Without Symbiodinium cell counts it’s difficult to validate this statement and the use of “significant”.
332: remove period before citations
356-358: Sentence should be edited to reflect the correct findings, as Closek (2014) identified more bacterial diversity in tissue from diseased colonies than non-diseased, but more bacterial diversity was noted in the transition tissue over the lesion tissue.
377: “diseases” should be diseased
Figure 1: Are these photos of the colonies sampled? If so, does the colony in Fig1B show any differences in the lesion, as the bands are morphologically different from the size of bands seen in A & C. If not, there should be a footnote indicating that the photos are representative of the YBD signs seen in the field.

---

## Round 0.2 · Minor Revisions

· Academic Editor

Minor Revisions

Please make sure you address the remaining minor criticisms brought up after the new round of reviews. They will help you improve the final version of your manuscript. Congratulations.

Reviewer 2 ·

Basic reporting

In the revised version of the manuscript, the authors worked on reorganizing the conclusions and making them more concise; they also moved methodology text to the appropriate section, and included information on the sampling. However, they ignored most of my comments on written English that have the purpose of presenting a clear proof-read text. They keep using colloquial English, for example on line 322 “…zoox densities are not necessarily lower…” when not even zooxanthellae would be the appropriate term. Or on lines 178 and 208, they refer to “Taq” (lines 178 and 208) instead of Taq DNA polymerase, when the reader can get confused with the Taq I endonuclease. Also, they named a REAL Norther blot, “virtual” (lines 229 and 241). Further, in Fig 1 “photos” should be photographs. Comments I made on the pdf, for example on line 426 Seneca et al (2010) that was not listed in the references, was not corrected; the reference on line 167, Hubank (1999) should be Hubank et al., and was not corrected. Several words that were mistyped in the original submission are still uncorrected. Please attend to the comments.

Experimental design

In the Results section, there is no mention of the results obtained with Poly A binding protein, although the expression of this gene was mentioned in the methods section as the chosen reference gene, since its expression was stable. I think it is important to show the expression results obtained with this gene and also with SOD-like gene as part of the results. I would suggest being consistent in the way all results are presented, and perhaps consider a graphic for both genes, even though there was not a significant variation in their expression.

Validity of the findings

Regarding this paragraph (lines 332-335) “Little attention has been given to the interaction of mitochrondria and chloroplast in stressed Symbiodinium. As the balance between carbon fixation and photorespiration fluctuates in favor of photorespiration in chloroplasts, photorespiration facilitates NADH production in mitochondria.” It is not clear from the text what is the connection between photorespiration and increased expression of cytochrome oxidase (cause/effect). Please clarify and include citations for photorespiration of symbionts in YBD affected corals if any. Since production of ROS does not seem to be a symptom of diseased colonies, and concurrently no increased expression of SOD was detected, photorespiration is not easily included in this scenario, as has been suggested to occur in green pants facing diverse abiotic stress (high light, low CO2, drought/water stress, etc.) as a means to maintain redox balance by preventing ROS accumulation (Voss et al 2013, as cited by the authors). You may want to check chlororespiration, rather.

The response given by the authors about my comment on the lack of reference genes for quantitating symbiont transcripts was convincing, in particular since the authors stated more clearly in the discussion/conclusions sections, that it is the expression changes in all three genes (two from the coral and one from the symbiont) that a disease/affected coral may show, and constitutes the “molecular fingerprint”. However, I remain skeptical of its reproducibility since, as I have stated before, it cannot be ignored that a coral involves at least two species of very different organisms, and even more critical, Orbicella faveolata presents more than one phylotype of Symbiodinum within their tissues. So I am not certain that three genes will behave the same in all diseased O. faveolata corals, and will diagnose correctly affected colonies. Input from the directly interested community will improve the usefulness of this biomarker.

Additional comments

I think that the manuscript, although improved overall, still needs revision.
Line 130 = “the objective of this study is…” should be changed to “the objective of this study was…”
Line 229 = I insist in the term “virtual Northern blot”; if it was actually performed why call it virtual???
Lines 236-238 = this phrase should be revised: “Comparisons of signal intensities between identical membranes probed with a different DIG-labeled amplicon determined which RDA products should be further analyzed.” If you mean that this protocol was actually used, then why further analysis?
Line 241 = 14 GOIs were screened to choose finally 5 potentially informative GOIs, but there is no mention on how the screening process was made.
Line 291 = “there were significantly differences in expression” please revise
Line 340 = “The zooxanthellae response for elevated energy demands…” please revise

Reviewer 3 ·

Basic reporting

No Comments

Experimental design

No Comments

Validity of the findings

No Comments

Additional comments

Overall, the manuscript has incorporated the majority of the reviewers' suggestions. There are still a few minor changes that need to be made, but upon those edits, I believe the manuscript should be accepted.

320-347: The cytochrome oxidase section in the Discussion should also recognize the possibility of other things contributing to the changes in expression, as fluctuations in other symbionts (i.e. bacteria) could also trigger this response. Otherwise, the arguments in this particular paragraph require further support.

327-328: Stating that a future comparison can be done seems out of place here and unnecessary.

Typo in Figure 3 “Disese” → “Disease”

---

## Round 0.3 · accepted · Accept

· Academic Editor

Accept

Congratulations on a much improved version of the manuscript.